# Development of Multi-Actor Multi-Criteria Analysis Based on the Weight of Stakeholder Involvement in the Assessment of Natural–Cultural Tourism Area Transportation Policies

**Heru Purboyo Hidayat Putro [1], Pradono Pradono [1] and Titus Hari Setiawan [2],***

[1] School of Architecture, Planning and Policy Development, Bandung Institute of Technology, Bandung 40132, Indonesia; herupur@pl.itb.ac.id (H.P.H.P.); pradono@pl.itb.ac.id (P.P.)
[2] Graduate Program in Transportation, School of Architecture Planning and Policy Development, Bandung Institute of Technology, Bandung 40132, Indonesia
* Correspondence: titusharisetiawan@gmail.com

**Abstract:** Multi-actor multi-criteria analysis (MAMCA) was developed with a process involving the participation of various stakeholders. Stakeholders express various criteria as measures for the achievement of their respective goals. In general, the assessment of each stakeholder is considered to have the same weight. In reality, the weight of each stakeholder's involvement in policy decision making is not the same. For example, the government's assessment weight will be different from those of local business actors. In this study, the authors developed a multi-actor multi-criteria analysis method by adding the weight of stakeholder involvement when making decisions about transportation policies that support sustainable mobility in protected natural–cultural tourism areas. The weight of involvement was developed through stakeholder participation. Stakeholders were asked to provide weights for all stakeholders other than themselves using the AHP method. The results of this weighting were then averaged and considered as the stakeholder assessment weights. Adding stakeholder weighting can also improve the quality of decisions by avoiding bias and following the principle of fairness in the assessment.

**Keywords:** multi-actor multi-criteria; stakeholder involvement; transport policy; natural–cultural tourism area

## 1. Introduction

Multi-criteria decision making (MCDM) has been widely used in various fields of study and for real-world problems and complex problems in the decision-making process [1]. MCDM is considered effective at unraveling issues that involve conflicts between parties [2] and that are difficult to measure with certainty [3]. MCDM is different from the decision-making (DM) approach, which was originally used for optimal problem solving [3], in that it applies quantitative surveys [4]. MCDM focuses on making decisions that are most likely to be carried out by many parties based on the problem's structure, taking all influential aspects into account [5]. Regarding its theoretical basis, MCDM was developed based on a systems approach. A systems approach is defined as a holistic view of a complex and interdisciplinary problem to achieve system goals [6]. MCDM is implemented with an understanding of the system under study, as well as the subsystems and their interrelationships, in order to achieve the objectives. Subsystem abstractions and interrelationships are actually expressed in the multi-criteria-based model; therefore, it is commonly called a multi-criteria decision model [7].

Research using human knowledge (expert knowledge) is quite useful in the field of transportation [8] and sustainability studies [9]. In the later development of MCDM, attention was paid to multiple actors [7], which was considered important in its evolution. This then became known as multi-actor multi-criteria analysis (MAMCA). In MAMCA, the

opinions of various stakeholders become input for the decision-making process. This is especially important for developing and densely populated countries such as Indonesia, where there are many stakeholders involved in transportation issues.

The involvement of stakeholders in decision making is participatory [10]. Their involvement can balance and integrate the various dimensions of sustainability [11]. It is recommended that stakeholder involvement be carried out in working groups [12]. However, in the field, this is not easy to implement. The weighting of criteria given by stakeholders uses an analytic hierarchy process (AHP) [13–16].

Stakeholders are people who have an interest in or an influence, financially or otherwise, on the impact of each decision made [17]. Stakeholder analysis is a tool for precisely identifying the various stakeholders that need to be involved and the views that should be considered in the evaluation process. In the scientific literature, several methods are described that can be used to obtain an appropriate list of stakeholders. Documentation along with legislative and administrative analysis, complemented by in-depth interviews with citizens and other interested parties, can create lists of the stakeholders involved [18]. After specific stakeholders have been identified, they are asked who should be involved. Policymakers must include all affected actors in the list of stakeholders [12,19]. Because their participation can potentially influence decisions, stakeholders must be involved throughout the planning and management processes [20]. As many stakeholders as possible should view the decision-making process as fair so that they will support the decision [20].

Stakeholder involvement in sustainability assessment is an important issue, because it can improve the quality of decision making, allow for the consideration of diverse values, promote social learning, and build trust [21,22]. In this study, in order to normalize the presence of stakeholders who feel powerful and have a lot of interest while others feel less important when, according to the other party, the interests are actually very important, additional analysis is carried out by weighting the stakeholders. In this method, each stakeholder assesses the weights of the other stakeholders. The tool used for this peer valuation is AHP. The goal of this study is to examine the possibility of using MAMCA by adding the step of stakeholder weighting in order to select alternative transportation policies for tourist destination areas that can meet environmental and cultural requirements. The hypothesis of this study is that stakeholder weighting will result in smaller standard deviations than if each stakeholder has the same weight as in the original MAMCA.

### 1.1. Stakeholder Perspectives in the Multi-Actor Multi-Criteria Analysis Method

Multi-actor multi-criteria analysis is a multi-criteria decision analysis method that has been widely used to solve transportation problems involving multidisciplinary and multi-stakeholder issues [23,24]. Decision making in the transportation sector is often fraught with intense contention because it can create advantages and disadvantages for various stakeholders. It is necessary to involve different perspectives from various stakeholders in the decision-making process to obtain sustainable solutions in the transportation sector [25]. Decision making through stakeholder participation is very important because it helps to identify and analyze the priorities of different stakeholders, increases decision acceptance rates, and strengthens the resilience and quality of the decisions [26].

The current multi-actor multi-criteria analysis method is conducted by selecting the criteria and weighting the stakeholder criteria, with the involvement of each stakeholder considered to be the same or 100%, even though the involvement of each stakeholder in the policy assessment is not the same. Other stakeholder-weighting approaches should also be analyzed [17]. This study extends multi-actor multi-criteria analysis (MAMCA) by adding the weight of stakeholders, using the potential of MAMCA to consider and involve stakeholders from the beginning to the end of the decision-making process [27]. In the study, each stakeholder assesses the other stakeholders according to the proportion of their involvement in making transportation policy decisions in protected natural–cultural tourism areas. Internal assessments produce criteria for each stakeholder.

Usually in MAMCA, as explained in [25], each stakeholder has equal weight. In this study, we conduct an external assessment of the involvement of stakeholders in making transportation policy decisions in protected natural–cultural tourism areas, which results in the weighting of stakeholder involvement. Stakeholder weighting is carried out by peer assessment and by using pairwise comparison. In this study, the general AHP was used instead of the interval AHP [28], considering that the evaluators comprised 20 stakeholders who could be grouped into six groups. These groups were representatives of the central government, provincial governments, local governments, local communities, and users and tourists. They manage or are related to four main problems: transportation, tourism, the environment, and historical heritage, which affect the welfare of local residents and actors providing transportation and tourism services. According to those people, these problems can be considered negotiable, so there may be no need to formulate a method or consider the existence of non-negotiable elements [29]. Stakeholders in the field are parties who continue to negotiate with each other so that their interests can be realized. Heterogeneity, or the presence of multiple stakeholders and multiple criteria, is an issue that needs to be considered when determining policies regarding sustainable tourism destinations [30]. The method of stakeholder participation in multi-actor multi-criteria analysis with the weighting of stakeholders to produce a transportation policy scenario was used for our case study.

### 1.2. Policy in Protected Tourism Areas

The impact of tourism on the sustainability of tourist destinations is questionable. Managers need to be aware of the various impacts of visitors and private vehicles, including increased air and noise pollution, damage to roadside vegetation, lack of parking spaces, visitor stress, traffic congestion, and climate change [31–37]. It is very important to pay attention to the mobility of tourists in destinations. Site destination managers must balance visitor mobility with conservation goals [38]. The right transportation policy is needed to support mobility in protected areas for sustainable natural–cultural tourism. Based on the literature on alternative transportation and tourism destination policies, this study forms a transportation policy scenario in protected areas for natural and cultural tourism.

#### 1.2.1. Incentives and Disincentives

Table 1 summarizes previous research related to transportation policy models in natural tourism areas.

**Table 1.** Transportation policies in natural tourism areas.

| No. | Authors | Alternatives/Scenario |
|-----|---------|------------------------|
| 1 | Pettebone et al., 2011 [39] | Scenarios: (1) use private vehicles, (2) use park and ride and shuttle, and (3) use shuttle from Estes Park |
| 2 | Taff, 2013 [40] | Alternative transport sistem |
| 3 | White, 2007 [41] | Alternative transport system |
| 4 | Mace, 2013 [42] | Mandatory alternative transportation system |
| 5 | Steiner and Bristow, 2000 [43] | Scenarios: (1) pay for road use and continue the journey, (2) park and transfer to alternative transport, and (3) go elsewhere |
| 6 | Beunen, 2008 [44] | Incentives include facilities at the gateway to attract visitors to park and leave their cars (concentrate traffic flow at gateway) |
| 7 | Regnerus et al., 2007 [45] | Parking and activity gateways, traffic flow, and visitor behavior |
| 8 | Holding and Kreutner, 1998 [46] | Carrot and stick: restrictions on private vehicles, public transportation, and park and ride |

Most stakeholders in natural tourism areas apply several transportation policies to support tourist mobility in the form of incentives and disincentives, such as the following:

- Car restrictions: Restrictions are imposed on private vehicles at popular destinations to avoid the accumulation of vehicles in parking spaces and traffic jams, accompanied by incentives to use public transport [35,45].
- Public transportation: Public transport is used in tourist areas as an incentive for tourists to not bring their private vehicles into protected tourism areas in order to preserve natural and cultural value. Public transportation is expected to bring tourists to attractive destinations in a balanced manner without environmental degradation. The use of public transportation can overcome the capacity constraints, congestion, and environmental impacts caused by tourism [32,47–51]. In some national parks and rural tourist destinations in Europe and the United States, public transport is a solution and part of the recreational tourism experience.
- Park and ride gateway management: Infrastructure is developed at the gateway to make a mode change, providing high-quality facilities to attract visitors to voluntarily transfer from their own vehicles to public transport [52].

### 1.2.2. Tourism Zoning

Destination boundaries support a conceptual framework in which tourist consumption patterns play a more fundamental role, such as having zoning determined by considering visitors rather than administrative areas [53]. A method for identifying tourism zones can be based on tourist consumption patterns and the time and distance between attractions, which is influenced by the spatial distribution of resources, including the distance to tourist objects and the intensity of their specificity and not by the administrative area [54,55]. Areas with greater concentrations of unique attractions have higher potential for attracting tourists. Based on this theory, the Dieng tourist area ignores administrative boundaries between two districts and is divided into zone A and zone B, based on the intensity of the visit and the specificity of the tourist attraction, as is shown in Figure 1. Zone A is the area with the most tourist attractions, as labeled in the figure: (1) Arjuna Temple, Gatotkaca Temple, and the Kaliasa Museum; (2) the Sikidang Crater; (3) the Dieng Plateau Theater; (4) Warna Lake; and (5) Welcome to Dieng. Zone B also has tourist attractions, but they are not in much demand, including Merdada Lake, the Sileri Crater, Jolotundo Well, and Mount Prau.

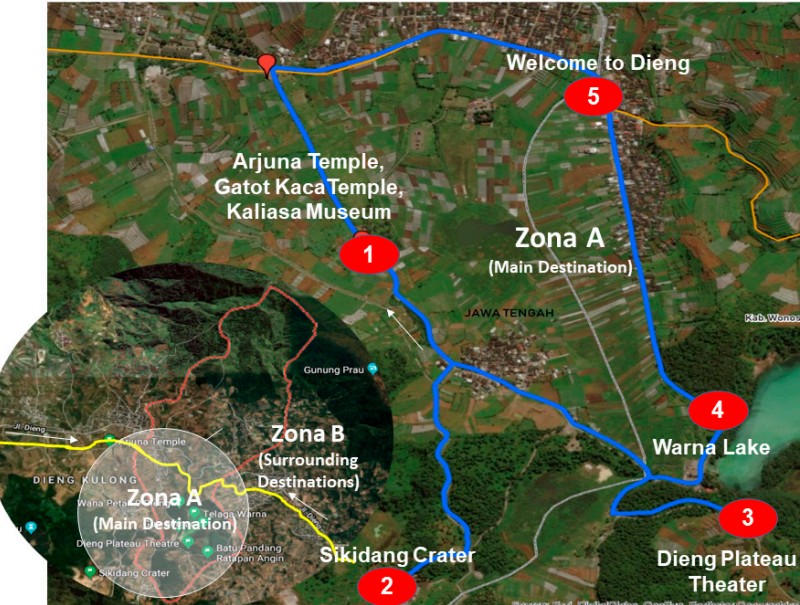

**Figure 1.** Dieng tourist area map.

## 2. Materials and Methods

In developing this methodology, the multi-criteria multi-actor analysis method was labeled A, and the multi-actor multi-criteria method with the weighting of stakeholder involvement was labeled B.

### 2.1. Steps in the Multi-Actor Multi-Criteria Analysis Method (A)

1.    Define alternative solutions to existing problems

At this stage, reasonable alternatives to transportation are compiled into a policy scenario according to the conditions of the study area. In this case, the tourist areas consisted of the main attractions of zone A and zone B, which were supporting attractions combined with the incentives and disincentives of transportation policy.

2.    Stakeholder analysis

At this stage, we selected the stakeholders who would be involved. This could be discovered through official documents, such as the duties and authorities of government and non-government organizations. Then, the snowball sampling technique was carried out on the target stakeholders.

3.    Criteria and weight definitions

We conducted stakeholder interviews regarding their criteria for determining transportation options in natural and cultural tourism areas and assessing the weight of each stakeholder's criteria.

4.    Criteria, indicators, and measurement methods

We looked for indicators from the criteria in the literature and in consultation with stakeholders. The indicators could be quantitative or qualitative if no measurement or data were required.

5.    Overall analysis and ranking

We conducted an overall assessment of the policy scenario, used the pairwise comparison method for each criterion, and then aggregated the results with the criteria weight for each stakeholder.

6.    Results

The overall results were obtained in the form of selected policy scenarios for each stakeholder and overall transportation policy scenarios in the natural–cultural tourism area (Figure 2) [26].

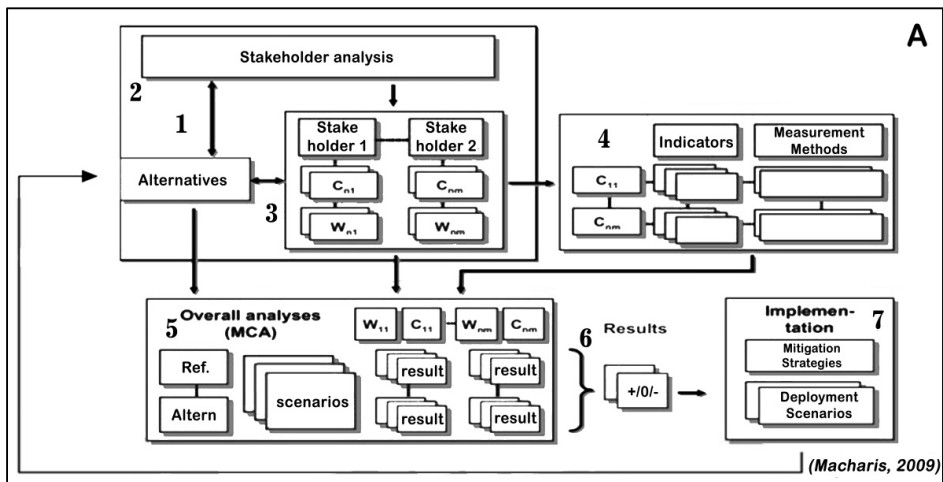

**Figure 2.** Multi-actor multi-criteria analysis (method A) [17].

*2.2. Multi-Actor Multi-Criteria Based on the Weight of Stakeholder Involvement (B)*

Stakeholders perform an overall assessment of the involvement of other stakeholders. Stakeholders raise their own criteria, and their involvement is assessed by the others (they do not judge themselves) to determine transportation policies that support protected areas of sustainable natural–cultural tourism (Figure 3).

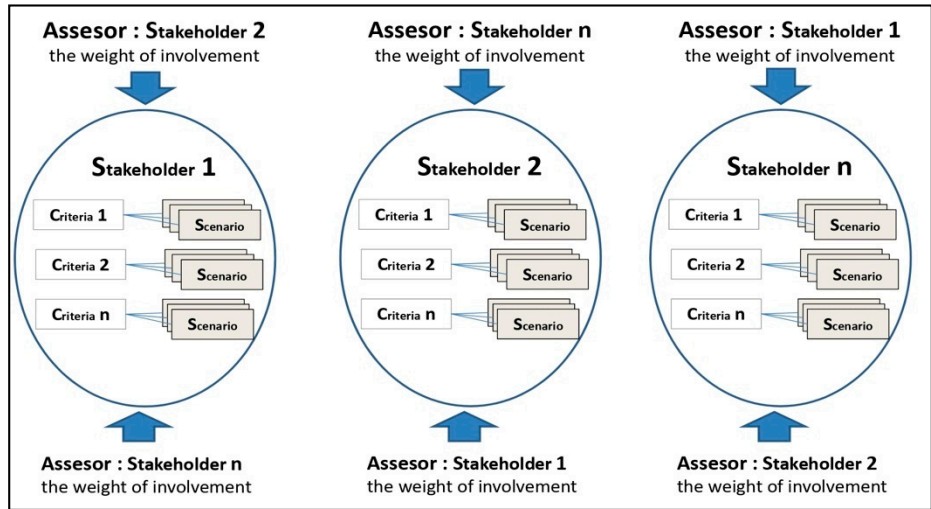

**Figure 3.** Weight of stakeholder involvement in a transportation policy scenario.

Multi-actor multi-criteria analysis based on stakeholder involvement differs from the previous method [17,26]. The steps are as follows:

1.  Determine alternative solutions to problems. These can also be combined into scenarios that suit the conditions and are supported by stakeholders.
2.  Conduct stakeholder analysis by selecting stakeholders to be interviewed using the snowball sampling technique, whereby each stakeholder is asked to provide recommendations for other stakeholders. Then, conduct interviews with the recommended stakeholders. This analysis should be viewed as an aid to identifying the interests of various stakeholders and views to be taken into account in the evaluation process. Based on the results of primary data collection, 20 respondents were selected by snowball sampling.
3.  Stakeholders are asked to provide criteria, according to their point of view, to measure transportation policies that support the sustainable mobility of tourist areas. Each stakeholder has the same criteria weight of 100%.
4.  Look for criteria indicators in the literature and selected by stakeholders, and then perform a pairwise comparison between the criteria using the Saaty scale, which produces the criteria weights. Measurement of this indicator can be qualitative or quantitative. This indicator is used as a tool to measure policy scenarios based on the criteria chosen by the stakeholders. Each stakeholder chooses their own criteria, and the total criterion weight of the stakeholder value is 100%.
5.  Conduct weighted assessment of stakeholder involvement. This is an expansion of the previous multi-actor multi-criteria method where each stakeholder has the same weight, so more research on the stakeholder's weight is needed [17]. In this study, the weight of the stakeholder's involvement is considered as each stakeholder's view of the other stakeholders who will influence the final assessment. This assessment can be made after snowball sampling locks in the results of the stakeholders involved as a whole. Each stakeholder conducts an assessment of other stakeholders but does not assess themself. In this assessment, 20 stakeholders were involved in assessing protected natural–cultural tourism areas.

6.   Assess scenarios using the pairwise comparison method for each criterion, and then multiply the results by the weight of the criteria and stakeholder involvement, resulting in the selected policy scenario.

There are significant differences in the expanded multi-actor multi-criteria analysis, specifically in assessing peer stakeholders, which ultimately becomes the stakeholder weight (stage 5) and the final construction of the policy scenario, using the weighting of criteria and stakeholder involvement (stage 6). The differences are shown in Figures 4 and 5.

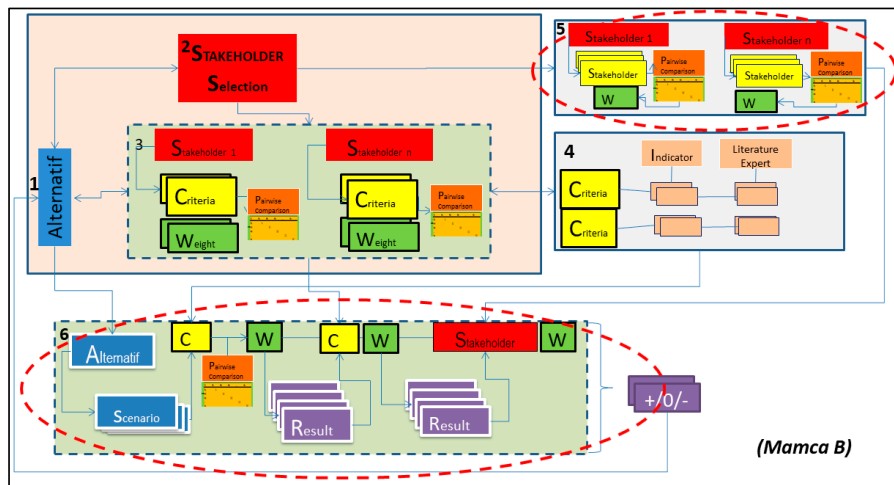

**Figure 4.** Multi-actor multi-criteria analysis based on the weight of stakeholder involvement (method B).

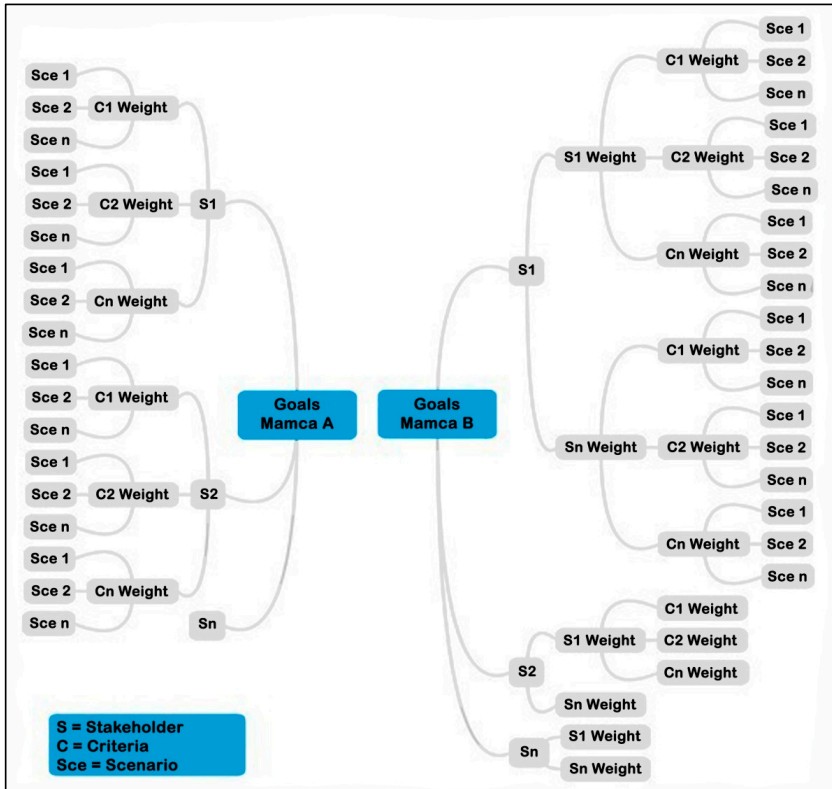

**Figure 5.** Differences in traditional MAMCA (A) and MAMCA with the stakeholder weight (B).

## 3. Results

### 3.1. Combining Alternatives into Scenarios

At this stage, the tourism zoning policy was combined with the incentives and disincentives policy to obtain a sustainable natural–cultural tourism area. Rigorous application of the intervention scenario would result in better sustainability but would have an impact on the economies of local communities and businesses. On the other hand, the absence of intervention would result in damage to the natural–cultural values that attract visitors. If the attraction is lost, it is feared that tourism would also be unsustainable. The mode of transportation used generally depends on the number of groups:

(1) Individual tourists use rural transportation modes from the boundary of zone B to the tourism area of zone A in Dieng. However, there is no special public transportation to explore interesting places in the tourist area.

(2) Large groups of tourists use large buses and are transferred to the boundary of zone B on small buses to explore destinations in zone A, with payments made using a rental system. The main reason for the transfer to small buses is narrow road access.

(3) Tourists using private vehicles or cars can go directly to zone A in the Dieng tourist area. Unfortunately, most tourists use cars to go to tourist areas, especially in zone A. This is a major problem, and at the peak of tourist visits or special events, tremendous traffic jams occur. In addition, heavy tourist activity results in environmental degradation. The issue of the sustainability of tourist destinations as protected areas with natural and cultural heritage value is also in question. Transportation policies to be used as guidelines for tourist mobility must encourage upholding these values.

This scenario is a combination of several alternative mobility policy solutions in natural and cultural tourism areas in the core in zone A and the surrounding area in zone B, with incentives and disincentives such as the provision of public transportation, park and ride, and car restrictions. The scenarios were made into four categories, from the least or no policy intervention to the greatest, and assessed by the stakeholders (Figure 6).

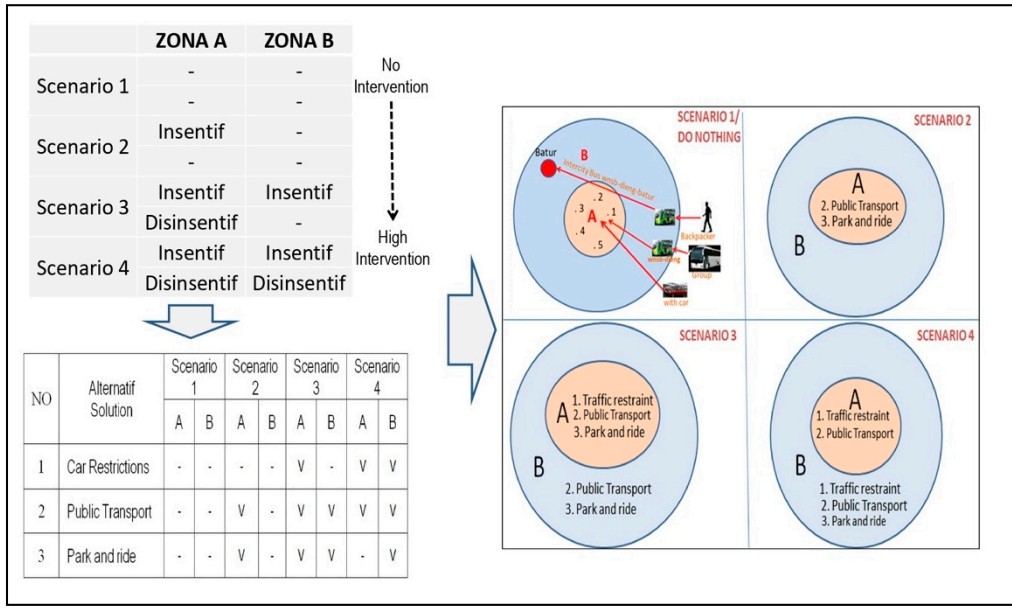

**Figure 6.** Transportation policy intervention scenario in the tourism zones.

Scenario 1 (No intervention)

This scenario represents the condition where there is no transportation policy intervention in the tourist area. Based on observations in the field, tourists visiting the Dieng tourist area visit zone A, which is the core with unique attractions that bring in visitors. This core area consists of Arjuna Temple, Gatotkaca Temple, and Kaliasa Museum in area 1;

Sikidang Crater in area 2; the Dieng Plateau Theater in area 3; Warna Lake in area 4; and Welcome to Dieng in area 5.

Scenario 2 (Little intervention)

This scenario involves an incentive policy intervention in zone A that provides public transport and park and ride facilities.

Scenario 3 (Moderate intervention)

In this scenario, there are incentives and disincentives in zone A in the form of car restrictions, park and ride facilities, and public transportation and only incentives in zone B, with the provision of public transportation and park and ride facilities.

Scenario 4 (Strong intervention)

In this scenario, there are incentives and disincentives in booth zones in the form of car restrictions and public transportation in zone A and car restrictions, public transport, and park and ride facilities in zone B.

### 3.2. Stakeholder Analysis

At this stage, the involved stakeholders were selected with snowball sampling. Each stakeholder interviewed was asked who should be involved. From the analysis results obtained, 20 stakeholders were selected (Table 2).

**Table 2.** Stakeholders related to transportation policies in natural and cultural tourism areas.

| NO | Institution | Duties and Authorities |
|---|---|---|
| S1 | Cultural Heritage Conservation Center of Central Java | National government is responsible for preserving cultural heritage in Central Java Province |
| S2 | Natural Resources Conservation Center of Central Java | National government is responsible for conserving natural resources in Dieng area, including several lakes |
| S3 | Regional Development Planning Agency of Central Java Province | Stakeholder of Central Java provincial government responsible for development planning |
| S4 | Central Java Provincial Transportation Office | Stakeholder of Central Java provincial government responsible for transportation sector |
| S5 | Central Java Provincial Tourism Office | Stakeholder of Central Java provincial government responsible for tourism sector |
| S6 | Central Java Land Transport Organization | Non-government stakeholders as an association of land transport companies |
| S7 | Regional Planning and Development Agency of Wonosobo Regency | Stakeholder of Wonosobo Regency government responsible for development planning |
| S8 | Transportation Office of Wonosobo Regency | Stakeholder of Wonosobo Regency government responsible for transportation sector |
| S9 | Tourism and Culture Office of Wonosobo Regency | Stakeholder of Wonosobo Regency government responsible for tourism sector |
| S10 | Wonosobo Land Transportation Organization | Non-government stakeholders as an association of land transport companies |
| S11 | Research and Development Planning Agency of Banjarnegara Regency | Stakeholder of Banjarnegara Regency government responsible for planning development research |
| S12 | Transportation Office of Banjarnegara Regency | Stakeholder of Banjarnegara Regency government responsible for transportation sector |
| S13 | Tourism and Culture Office of Banjarnegara Regency | Stakeholder of Banjarnegara Regency government responsible for tourism sector |
| S14 | Land Transportation Organization Banjarnegara | Non-government stakeholder as an association of land transport companies |
| S15 | Dieng Pandhawa (tourism awareness group) | Non-government stakeholder as local community that drives tourism sector in Dieng Wetan Village |

| NO | Institution | Duties and Authorities |
|---|---|---|
| S16 | Dieng Kulon (tourism awareness group) | Non-government stakeholder as local community that drives tourism in Dieng Kulon Village |
| S17 | Homestay association | Non-government stakeholder as local community of homestay businesses in Dieng |
| S18 | Association of Indonesian Tours and Travel Agencies (ASITA) | Non-government stakeholder as association of tours and travel agencies |
| S19 | Insan Pariwisata Indonesia (ISP) | Non-government stakeholder as community of tourism actors |
| S20 | Association of Indonesian Tours and Travel Agencies (ASPPI) | Non-government stakeholder as association of tourism actors |

### 3.3. Defining Criteria and Weights

Based on the selection and assessment of criteria of the stakeholders, 13 criteria were obtained, which were then coded and weighted using the pairwise comparison method as follows: C1, integrated transport system; C2, safety and security; C3, accessibility; C4, various transport systems; C5, comprehensive planning; C6, protection of cultural assets; C7, low-impact transportation; C8, visitor management; C9, support for local entrepreneurs; C10, visitor experience; C11, transport operational efficiency; C12, transport equality; and C13, support for cultural events (Table 3).

### 3.4. Weighting of Stakeholder Involvement

Based on the assessment of all stakeholders involved in determining the transportation policy in the natural–cultural tourism area (Table 4), the average stakeholder weight from highest to lowest was as follows: the Central Java Province Transportation Agency (S4 = 9.9%), Regional Development Planning Agency of Central Java Province (S3 = 7.70%), Central Java Provincial Tourism Office (S5 = 7.28%), Transportation Office of Banjarnegara Regency (S12 = 6.57%), Transportation Office of Wonosobo Regency (S8 = 6.44%), Tourism and Culture Office of Wonosobo Regency (S9 = 6.43%), Tourism and Culture Office of Banjarnegara Regency (S13 = 5.92%), Wonosobo Land Transportation Organization (S10 = 5.74%), Land Transportation Organization Banjarnegara (S14 = 5.42%), Regional Planning and Development Agency of Wonosobo Regency (S7 = 4.66%), Cultural Heritage Conservation Center of Central Java (S1 = 4.62%), Research and Development Planning Agency of Banjarnegara Regency (S11 = 4.47%), Central Java Land Transport Organization (S6 = 4.28%), Dieng Kulon Tourism Awareness Group (S16 = 4.14%), Natural Resources Conservation Center of Central Java (S2 = 3.61%), Dieng Pandhawa Tourism Awareness Group (S15 = 3.58%), homestay association (S17 = 2.76%), ASITA (S18 = 2.46%), IPI (S19 = 2.31%), and ASPPI (S20 = 1.72%).

**Table 3.** Results of stakeholder selection and assessment of criteria.

| Criteria | National | | Central Java Province | | | | Wonosobo Regency | | | | Banjarnegara Regency | | | | Local Community | | | | User | |
|---|---|---|---|---|---|---|---|---|---|---|---|---|---|---|---|---|---|---|---|---|
| | S1 | S2 | S3 | S4 | S5 | S6 | S7 | S8 | S9 | S10 | S11 | S12 | S13 | S14 | S15 | S16 | S17 | S18 | S19 | S20 |
| C1 | 3.9% | 4.0% | 19.5% | 8.7% | 22.0% | 13.1% | 8.8% | 5.3% | 26.6% | 49.8% | 14.1% | 7.4% | 8.3% | 8.2% | 3.2% | 3.4% | 35.3% | 9.4% | 4.4% | 26.3% |
| C2 | - | - | - | 26,6% | - | 54.0% | - | 28.2% | - | - | 26.1% | 26.1% | - | 25.7% | 25.0% | 41.9% | - | 41.0% | 27.2% | - |
| C3 | - | - | 13.3% | 12.4% | - | - | 16.0% | 17.0% | - | - | 7.4% | 14.1% | - | - | 5.0% | 5.2% | 17.2% | 5.5% | 9.0% | - |
| C4 | - | - | 6.7% | 5.4% | 5.3% | 5.7% | 19.7% | - | 3.9% | 4.9% | 48.3% | - | 4.2% | 4.2% | - | - | 7.1% | 3.1% | - | 5.7% |
| C5 | 28.1% | 48.7% | 44.4% | 46.9% | 38.4% | 27.2% | 55.5% | 28.5% | 17.4% | 25.1% | - | 48.3% | 46.5% | 47,9% | - | 14.7% | - | 24.8% | 44.9% | 12.2% |
| C6 | 7.0% | 13.0% | - | - | - | - | - | - | - | - | - | - | - | - | - | - | - | - | - | - |
| C7 | 46.4% | 26.1% | - | - | - | - | - | - | - | - | - | - | - | - | - | - | - | - | - | - |
| C8 | 14.6% | 8.2% | 4.6% | - | 21.7% | - | - | 21.1% | 34.3% | 13.1% | 4.0% | 4.0% | 26.4% | - | - | - | - | - | 14.6% | 55.8% |
| C9 | - | - | - | - | - | - | - | - | - | - | - | - | - | - | 43.3% | 26.4% | 40.4% | - | - | - |
| C10 | - | - | - | - | - | - | - | - | 7.4% | - | - | - | - | - | - | - | - | 16.2% | - | - |
| C11 | - | - | - | - | - | - | - | - | - | 7.1% | - | - | - | 13.9% | - | - | - | - | - | - |
| C12 | - | - | 11.6% | - | - | - | - | - | - | - | - | - | - | - | - | - | - | - | - | - |
| C13 | - | - | - | - | 12.6% | - | - | - | 10.4% | - | - | - | 14.6% | - | 8.6% | 8.4% | - | - | - | - |

**Table 4.** Stakeholder weights.

| Rank | Stakeholder | National | | | Central Java Province | | | Wonosobo Regency | | | | Banjarnegara Regency | | | | Local Community | | User/Tourist | | | | Sum | Average |
|---|---|---|---|---|---|---|---|---|---|---|---|---|---|---|---|---|---|---|---|---|---|---|---|
| | | S1 | S2 | S3 | S4 | S5 | S6 | S7 | S8 | S9 | S10 | S11 | S12 | S13 | S14 | S15 | S16 | S17 | S18 | S19 | S20 | | |
| 11 | S1 | | 1.2% | 2.5% | 1.3% | 2.3% | 3.00% | 2.1% | 8.2% | 1.4% | 3.6% | 10.2% | 7.6% | 1.7% | 6.0% | 10.0% | 12.6% | 14.6% | 1.4% | 1.0% | 1.7% | 92% | 4.62% |
| 15 | S2 | 2.3% | | 2.3% | 1.0% | 2.9% | 2.28% | 3.8% | 6.2% | 1.8% | 2.5% | 3.3% | 4.1% | 1.6% | 3.0% | 8.2% | 10.0% | 12.0% | 1.3% | 1.6% | 2.1% | 72% | 3.61% |
| 2 | S3 | 5.8% | 7.9% | | 1.2% | 12.61% | 1.56% | 4.1% | 13.6% | 12.6% | 1.9% | 11.0% | 13.6% | 12.6% | 7.5% | 10.8% | 13.6% | 11.1% | 9.2% | 1.2% | 2.0% | 154% | 7.70% |
| 1 | S4 | 10.0% | 14.2% | 13.2% | | 12.60% | 13.95% | 3.2% | 12.6% | 13.6% | 11.7% | 13.7% | 12.6% | 13.5% | 10.7% | 13.6% | 10.8% | 9.3% | 3.3% | 2.1% | 3.1% | 198% | 9.90% |
| 3 | S5 | 8.3% | 1.6% | 3.3% | 7.5% | | 10.17% | 5.6% | 10.9% | 10.1% | 14.2% | 12.7% | 10.0% | 10.0% | 8.0% | 12.6% | 7.8% | 7.0% | 2.7% | 1.7% | 1.5% | 146% | 7.28% |
| 13 | S6 | 1.0% | 10.1% | 8.1% | 1.5% | 7.5% | | 1.6% | 1.3% | 8.3% | 4.8% | 3.1% | 10.8% | 7.6% | 3.2% | 2.9% | 4.1% | 3.4% | 1.8% | 3.1% | 1.2% | 86% | 4.28% |
| 10 | S7 | 3.3% | 7.1% | 7.2% | 4.4% | 7.4% | 3.97% | | 10.1% | 5.8% | 1.0% | 4.4% | 4.4% | 4.3% | 5.5% | 1.7% | 4.4% | 1.8% | 4.8% | 7.6% | 4.0% | 93% | 4.66% |
| 5 | S8 | 7.7% | 5.0% | 13.0% | 3.1% | 11.0% | 4.11% | 8.3% | | 10.9% | 10.8% | 7.9% | 3.1% | 6.0% | 9.9% | 2.3% | 2.3% | 2.6% | 7.0% | 8.2% | 5.5% | 129% | 6.44% |
| 6 | S9 | 10.4% | 2.7% | 3.3% | 10.9% | 4.1% | 9.77% | 13.1% | 7.7% | | 6.0% | 5.9% | 2.9% | 3.3% | 4.2% | 3.1% | 3.1% | 2.5% | 11.4% | 12.6% | 11.9% | 129% | 6.43% |
| 8 | S10 | 4.5% | 6.9% | 10.6% | 1.6% | 7.2% | 5.69% | 13.2% | 3.8% | 4.3% | | 2.3% | 2.3% | 4.1% | 1.3% | 2.1% | 2.1% | 1.9% | 14.5% | 13.6% | 12.8% | 115% | 5.74% |
| 12 | S11 | 1.7% | 1.1% | 4.4% | 5.9% | 5.7% | 7.36% | 1.6% | 5.7% | 6.2% | 1.1% | | 8.2% | 10.8% | 1.5% | 4.1% | 7.7% | 3.7% | 1.1% | 1.3% | 10.0% | 89% | 4.47% |
| 4 | S12 | 13.0% | 9.2% | 10.5% | 3.0% | 10.6% | 0.95% | 1.3% | 4.4% | 7.7% | 8.4% | 8.5% | | 8.2% | 13.5% | 4.4% | 5.6% | 5.1% | 3.5% | 6.0% | 7.5% | 131% | 6.57% |
| 7 | S13 | 13.5% | 3.7% | 4.5% | 10.3% | 3.2% | 7.93% | 2.1% | 3.1% | 3.2% | 13.2% | 6.4% | 6.0% | | 1.1% | 5.6% | 6.0% | 4.7% | 8.5% | 10.0% | 5.3% | 118% | 5.92% |
| 9 | S14 | 5.8% | 12.3% | 8.7% | 2.2% | 4.2% | 12.45% | 1.2% | 3.3% | 4.6% | 6.4% | 2.4% | 5.6% | 5.6% | | 1.6% | 2.9% | 1.5% | 11.3% | 10.8% | 5.7% | 108% | 5.42% |
| 16 | S15 | 4.2% | 3.8% | 1.1% | 12.6% | 1.3% | 5.30% | 11.0% | 2.2% | 1.2% | 4.4% | 1.8% | 2.1% | 1.3% | 3.2% | | 1.7% | 8.7% | 1.8% | 2.9% | 1.0% | 72% | 3.58% |
| 14 | S16 | 3.1% | 2.1% | 1.1% | 12.9% | 1.2% | 3.89% | 10.2% | 1.5% | 1.0% | 3.2% | 1.7% | 1.6% | 1.2% | 12.5% | 7.6% | | 6.5% | 3.3% | 5.6% | 2.7% | 83% | 4.14% |
| 17 | S17 | 1.1% | 5.2% | 1.3% | 8.1% | 1.0% | 2.91% | 7.7% | 1.7% | 1.1% | 1.7% | 1.4% | 1.5% | 1.0% | 3.9% | 6.0% | 1.6% | | 2.4% | 4.1% | 1.6% | 55% | 2.76% |
| 18 | S18 | 1.3% | 2.7% | 1.8% | 5.5% | 2.1% | 1.52% | 3.9% | 1.3% | 2.4% | 1.4% | 1.3% | 1.3% | 2.9% | 1.0% | 1.2% | 1.3% | 1.3% | | 4.4% | 10.5% | 49% | 2.46% |
| 19 | S19 | 1.6% | 1.9% | 1.7% | 3.9% | 1.7% | 1.20% | 2.9% | 1.2% | 2.2% | 2.3% | 1.1% | 1.2% | 2.3% | 1.2% | 1.3% | 1.3% | 1.2% | 6.2% | | 9.7% | 46% | 2.31% |
| 20 | S20 | 1.2% | 1.4% | 1.3% | 2.9% | 1.5% | 2.00% | 2.9% | 1.1% | 1.6% | 1.3% | 1.0% | 1.1% | 2.1% | 3.0% | 1.0% | 1.0% | 1.1% | 4.5% | 2.3% | | 34% | 1.72% |
| | | 1.0 | 1.0 | 1.0 | 1.0 | 1.0 | 1.0 | 1.0 | 1.0 | 1.0 | 1.0 | 1.0 | 1.0 | 1.0 | 1.0 | 1.0 | 1.0 | 1.0 | 1.0 | 1.0 | 1.0 | 20.0 | 1.0 |

*3.5. Results of Assessment with MAMCA (A) and MAMCA with Stakeholder Involvement (B)*

The results of the assessment using MAMCA (method A) and MAMCA with stakeholder weights (method B) can be seen in Table 5, and the graphs can be seen in Figures 7 and 8.

**Table 5.** Differences in results of MAMCA (A) and MAMCA with stakeholder weights (B).

| Stakeholders | Sce 1 A | Sce 1 B | Sce 2 A | Sce 2 B | Sce 3 A | Sce 3 B | Sce 4 A | Sce 4 B |
|---|---|---|---|---|---|---|---|---|
| **S1** | 5.79% | 7.29% | 12.50% | 15.85% | 28.42% | 36.61% | 53.28% | 40.24% |
| **S2** | 4.87% | 9.25% | 26.90% | 15.35% | 14.24% | 35.42% | 53.99% | 39.98% |
| **S3** | 6.27% | 6.71% | 13.86% | 15.32% | 35.64% | 36.71% | 44.23% | 41.25% |
| **S4** | 5.75% | 13.15% | 14.24% | 18.50% | 33.01% | 36.08% | 46.99% | 32.26% |
| **S5** | 5.79% | 6.65% | 12.43% | 15.32% | 51.62% | 36.45% | 30.16% | 41.58% |
| **S6** | 5.74% | 8.47% | 13.58% | 15.67% | 45.25% | 35.96% | 35.43% | 39.90% |
| **S7** | 6.61% | 12.34% | 13.87% | 17.89% | 48.85% | 32.98% | 30.68% | 36.80% |
| **S8** | 5.63% | 7.17% | 20.45% | 14.97% | 43.07% | 35.92% | 30.85% | 41.95% |
| **S9** | 4.45% | 6.74% | 10.41% | 15.19% | 28.17% | 38.85% | 56.98% | 39.21% |
| **S10** | 5.89% | 7.68% | 12.76% | 16.26% | 28.47% | 38.07% | 52.88% | 37.99% |
| **S11** | 5.89% | 6.91% | 14.81% | 15.22% | 44.27% | 36.67% | 35.03% | 41.20% |
| **S12** | 5.72% | 7.10% | 13.62% | 15.22% | 32.26% | 37.67% | 48.39% | 40.01% |
| **S13** | 5.82% | 6.68% | 19.88% | 14.58% | 43.93% | 38.18% | 30.36% | 40.55% |
| **S14** | 5.70% | 9.15% | 12.23% | 17.43% | 30.07% | 36.26% | 52.00% | 37.15% |
| **S15** | 27.35% | 9.11% | 21.71% | 17.07% | 31.37% | 34.57% | 19.57% | 39.26% |
| **S16** | 12.42% | 6.87% | 30.28% | 15.65% | 36.44% | 34.91% | 20.86% | 42.57% |
| **S17** | 53.35% | 8.02% | 27.47% | 17.25% | 12.01% | 33.70% | 7.16% | 41.04% |
| **S18** | 5.73% | 7.49% | 13.70% | 15.21% | 37.47% | 35.45% | 43.10% | 41.85% |
| **S19** | 6.57% | 8.60% | 14.47% | 16.08% | 40.31% | 34.97% | 38.65% | 40.35% |
| **S20** | 6.17% | 6.86% | 12.76% | 14.99% | 50.83% | 35.46% | 30.25% | 42.69% |
| **Average** | 9.58% | 8.11% | 16.60% | 15.95% | 35.78% | 36.04% | 38.04% | 39.89% |
| **Standard Deviation** | 11.44% | 1.82% | 5.80% | 1.10% | 10.81% | 1.47% | 13.39% | 2.42% |

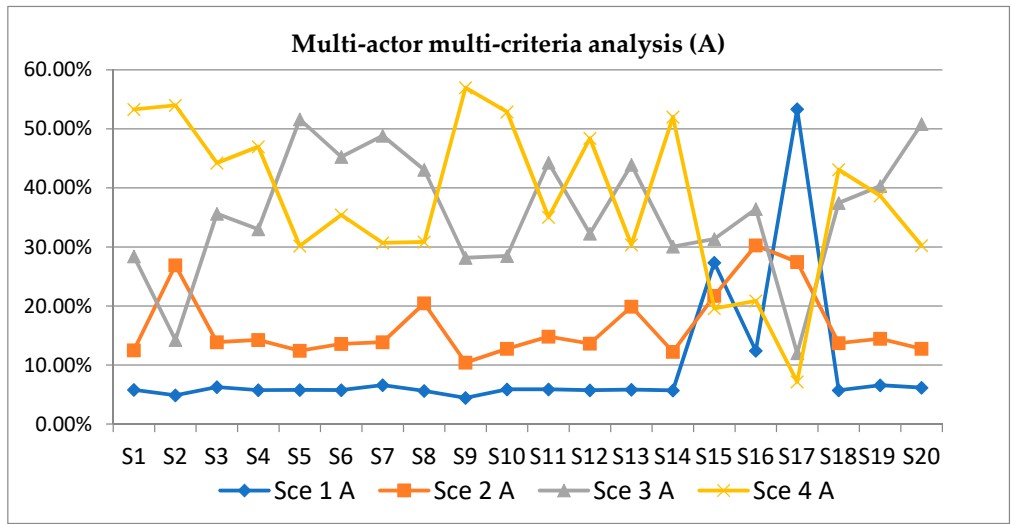

**Figure 7.** Results of the multi-actor multi-criteria analysis (method A).

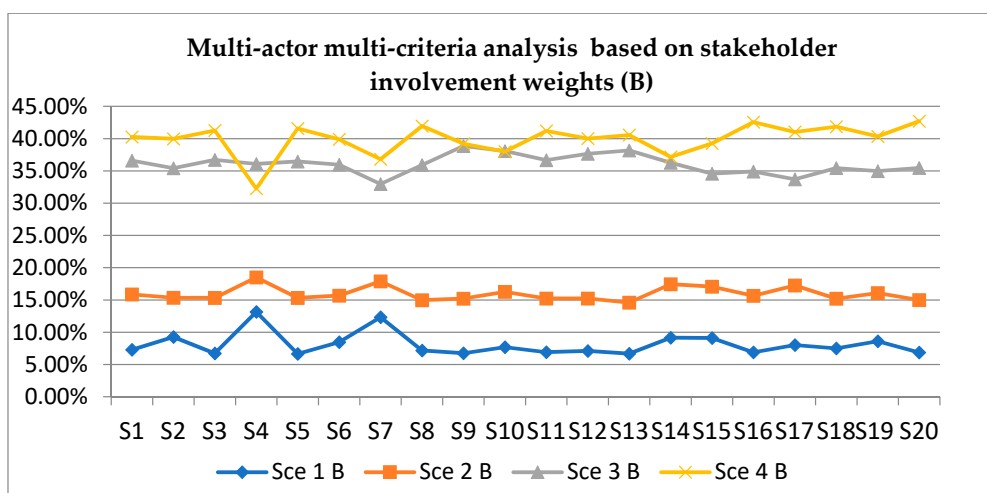

**Figure 8.** Results of multi-actor multi-criteria analysis with stakeholder weights (method B).

Figures 7 and 8 show the results of MAMCA analysis (method A) and MAMCA analysis with the weight of stakeholder involvement (method B). The assessment results by method B had a smaller standard deviation, while those by method A had a larger standard deviation. Next, we compare each scenario with the two methods.

## 4. Discussion

From the research using MAMCA (A) and MAMCA with the weight of stakeholder involvement (B), we could compare the results in each scenario as follows.

Figure 9 shows that in scenario 1, with method A, the Tourism and Culture Office of Wonosobo Regency (S9) had the lowest score, and the homestay association (S17) had the highest score, with a value distribution of 4.45–53.35% and standard deviation of 11.44%. With method B, the Central Java Provincial Tourism Office (S5) received the lowest score, and the Central Java Provincial Transportation Office (S4) received the highest score, with a value range of 6.65–13.15% and standard deviation 1.82%.

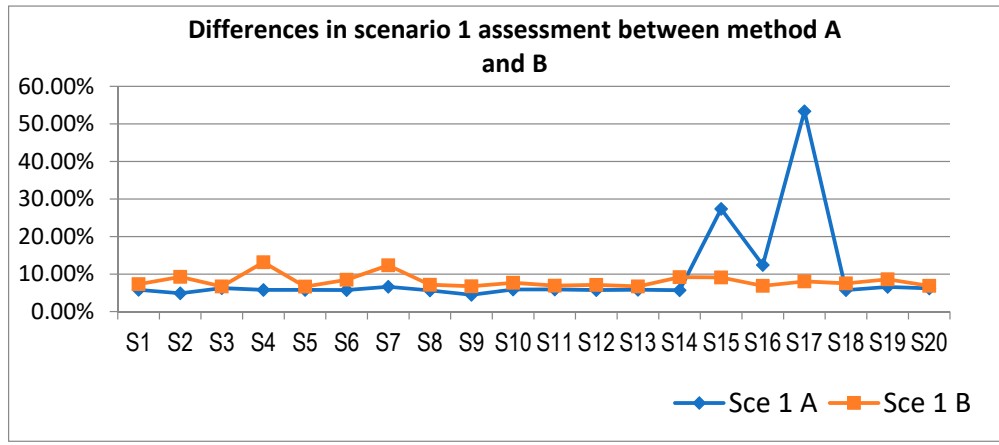

**Figure 9.** Differences in scenario 1 between multi-actor multi-criteria analysis (method A) and multi-actor multi-criteria analysis with the weight of stakeholder involvement (method B).

Figure 10 shows that in scenario 2, with method A, the Wonosobo Regency Tourism and Culture Office (S9) had the lowest score, and the Dieng Kulon tourism awareness group (S16) had the highest score, with a value distribution of 10.41–30.28% and standard deviation of 5.80%. With method B, the Tourism and Culture Office of Banjarnegara Regency (S13) received the lowest score, and the Transportation Office of Central Java

Province (S4) received the highest score, with a value range of 14.58–18.50% and standard deviation of 1.10%.

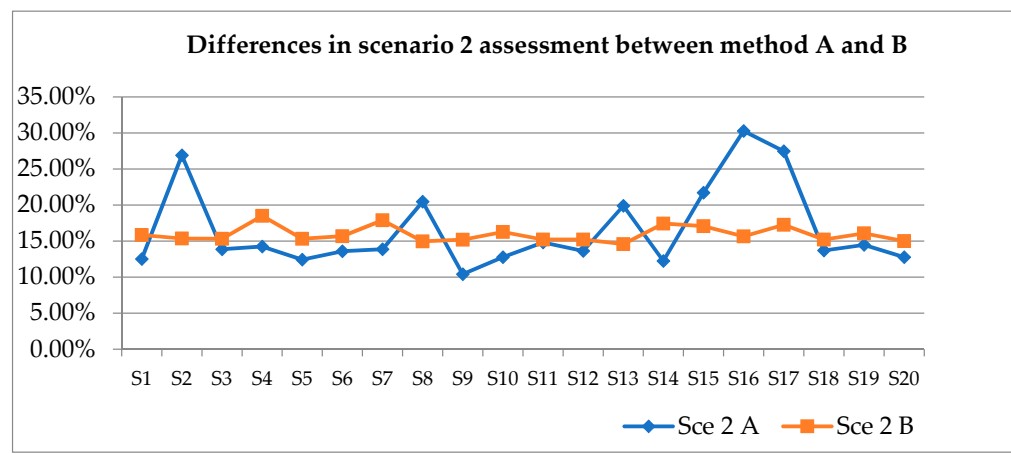

**Figure 10.** Differences in scenario 2 between multi-actor multi-criteria analysis (method A) and multi-actor multi-criteria analysis with the weight of stakeholder involvement (method B).

Figure 11 shows that in scenario 3, with method A, the homestay association (S17) had the lowest value, and the Central Java Provincial Tourism Office (S5) had the highest value, with a value distribution of 12.01–51.62% and standard deviation of 10.81%. With method B, the Regional Planning and Development Agency of Wonosobo Regency (S7) had the lowest score, and the Tourism and Culture Office of Wonosobo Regency (S9) had the highest score, with a value range of 32.98–38.85% and standard deviation of 1.47%.

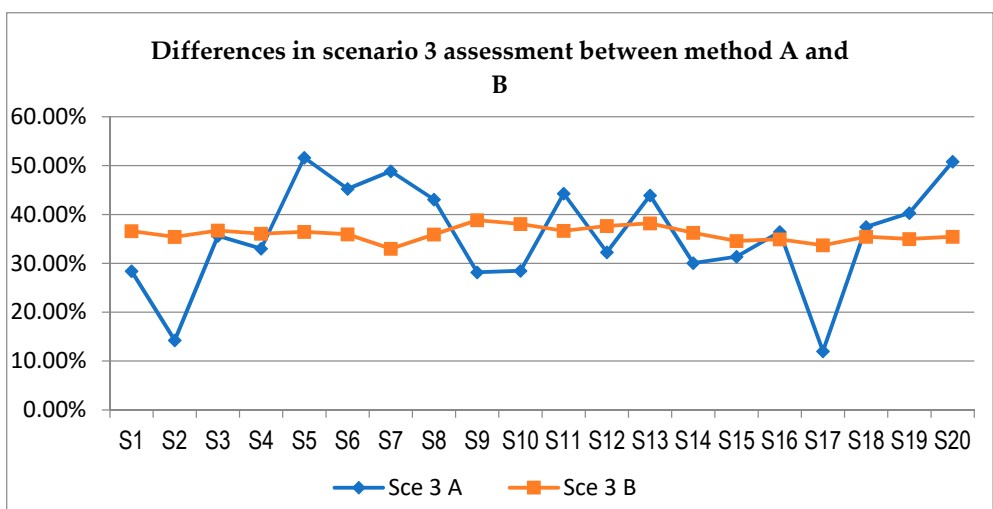

**Figure 11.** Differences in scenario 3 assessment between multi-actor multi-criteria analysis (method A) and multi-actor multi-criteria analysis with the weight of stakeholder involvement (method B).

Figure 12 shows that in scenario 4, with method A, the homestay association (S17) had the lowest score, and the Tourism and Culture Office of Wonosobo Regency (S9) had the highest score, with a value distribution of 7.65–56.98% and standard deviation of 13.39%. With method B, the Transportation Office of Central Java Province (S4) received the lowest score, and the ASPPI (S20) received the highest score, with a value range of 32.26–42.69% and a standard deviation of 2.39%. The final results of the assessment of the scenarios using methods A and B are shown in Figure 13.

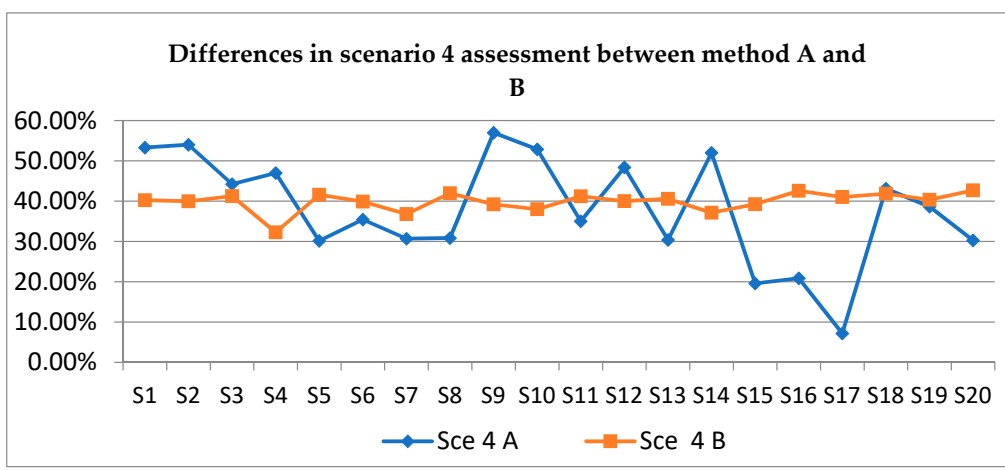

**Figure 12.** Differences in scenario 4 between multi-actor multi-criteria analysis (method A) and multi-actor multi-criteria analysis with the weight of stakeholder involvement (method B).

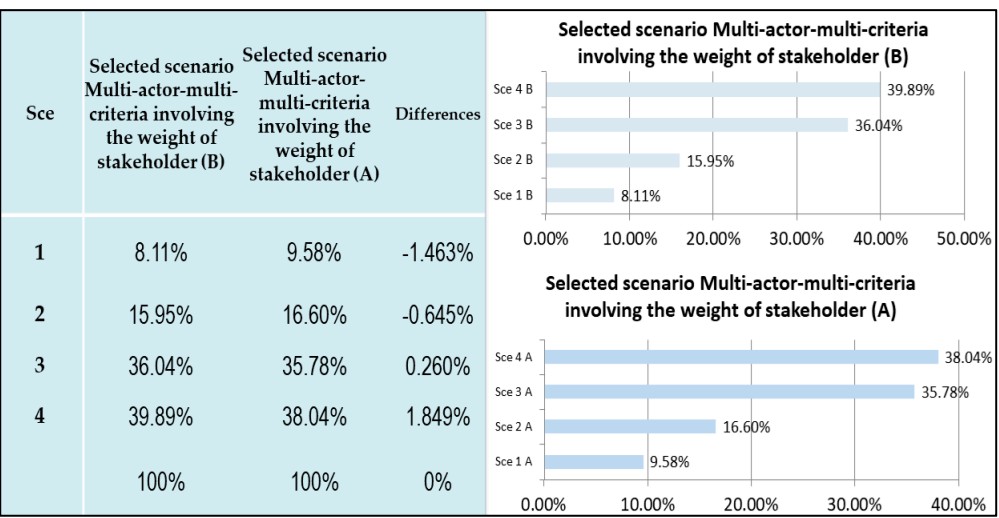

**Figure 13.** Results of the assessment of multi-actor multi-criteria analysis (method A) and multi-actor multi-criteria analysis with weight of stakeholder involvement (method B).

Based on the results of the analysis using method B, it was found that the transportation policy scenario chosen in the protected natural–cultural tourism area was scenario 4, with a weight of 39.89%, followed by scenario 3 with 36.04%, scenario 2 with 15.95%, and scenario 1 with 8.11%. The results of the assessment using method A showed scenario 4 with a weight of 38.04%, followed by scenario 3 with 35.78%, scenario 2 with 16.60%, and scenario 1 with 9.58%.

The assessment of the stakeholder as a form of support between stakeholders provided consistency in the assessment. However, as mentioned earlier, method B, which used stakeholder weights, had a smaller standard deviation than method A. This means that the difference in assessments among the stakeholders in B was smaller than in A. This was considered to have an impact on the decisions made, which would be more accepted by all parties because all stakeholders were asked to assess the weight of the interests of the others regarding their common problem. Consequently, it would be more sustainable. This study also found significant differences in the assessment criteria between local communities and other stakeholders (Tables A1–A6 in Appendix A). Local communities, as stakeholders who are directly affected, have a strong desire to keep their interests undisturbed. They (the local tourism actors) generally choose the do-nothing scenario. Perhaps they are concerned that other scenarios will cost them. This can be an indication of the need to disseminate



information about the danger of excess transportation in environmentally fragile areas. At the same time, it is necessary to carry out capacity building so that local stakeholders can develop tourism products that are more attractive to visit by using public transportation and ensure that the options in tourist areas, such as local public transportation and park and ride facilities, can still encourage tourists to enjoy their activities.

## 5. Conclusions

This paper applied multi-actor multi-criteria analysis (MAMCA) for the assessment of transportation policies in natural–cultural tourism areas in Indonesia as a case study. The analysis involved the participation of various stakeholders. Multi-actor multi-criteria analysis considering the weight of stakeholders expands the development of the original method with internal assessments in the form of stakeholder criteria and external assessments in the form of stakeholder weights. Proportional weighting of stakeholders can address variations in weights, as indicated by the smaller standard deviation in the implementation results, thus meeting the goal of this study: the addition of a stakeholder weighting step provided better results than the original MAMCA.

It should be noted that the MAMCA process becomes longer because the assessment of stakeholder weights can only be carried out after determining all of the stakeholders involved. It relies on peer-to-peer assessments, which makes the method more objective and avoids judgment bias by the researcher. By using stakeholder weights, the study results show that the variation in assessments among stakeholders is small. This verifies that the method is feasible to use even though it takes a longer time. The Supplementary Materials are available in Tables S1–S22.

Further studies can be carried out by conducting focus groups involving all stakeholders, where there can be discussions and debates about their common problems. Thus, it is hoped that stakeholders will have a similar understanding of the problems they may face related to the study area.

**Supplementary Materials:** The following are available online at https://www.mdpi.com/article/10.3390/a14070217/s1. Table S1: Cultural Heritage Conservation Center of Central Java, Table S2: Natural Resources Conservation Center of Central Java, Table S3: Regional Development Planning Agency of Central Java Province, Table S4: Provincial Transportation Agency Central Java, Table S5: The Central Java Provincial Tourism Office (Dispar Jateng), Table S6: The Central Java Land Transport Organization, Table S7: Regional Planning and Development Agency of Wonosobo Regency, Table S8: Transportation Office of Wonosobo Regency, Table S9: Tourism and Culture Office of Wonosobo Regency, Table S10: Wonosobo Land Transportation Organization, Table S11: Research and Development Planning Agency of Banjarnegara Regency, Table S12: Transportation Office of Banjarnegara Regency, Table S13: Tourism and Culture Office of Banjarnegara Regency, Table S14: Land Transportation Organization Banjarnegara, Table S15: Pokdarwis Dieng Pandhawa (Tourism Awareness Group), Table S16: Pokdarwis Dieng Kulon (Tourism Awareness Group), Table S17: Homestay association, Table S18: ASITA (Association of The Indonesian Tours and Travel Agencies), Table S19: IPI, Table S20: ASPPI, Table S21: Stakeholder Weight, Table S22: Multiplication of the Mamca(A) policy scenario matrix and stakeholder weight matrix.

**Author Contributions:** Conceptualization, H.P.H.P.; methodology, H.P.H.P. and P.P.; validation, P.P.; writing—original draft preparation, H.P.H.P. and T.H.S.; writing—review and editing, H.P.H.P., P.P., and T.H.S.; supervision, H.P.H.P. and P.P. All authors have read and agreed to the published version of the manuscript.

**Funding:** This research received no external funding.

**Data Availability Statement:** Not applicable; this study does not report any data.

**Acknowledgments:** This paper is part of a research road map within the System of Infrastructure and Transportation Research Group SAPPD ITB. The authors express their deepest gratitude to the Central Java Provincial Transportation and Tourism-related offices for the valuable data and information needed for this research. The content is solely the responsibility of the authors.

**Conflicts of Interest:** The authors declare no conflict of interest.

## Appendix A

**Table A1.** Results of pairwise comparison of the Saaty scale (from −9 to 9) for the national group.

| NO | S1 | | | S2 | | | Average | | |
|---|---|---|---|---|---|---|---|---|---|
| | SW | CW | Sce W | SW | CW | Sce W | SW | CW | Sce W |
| 1 | 0.0% | 3.9% | 5.79% | 1.22% | 4.0% | 4.87% | 0.61% | 3.95% | 5.33% |
| 2 | 2.3% | 0.0% | 12.50% | 0.00% | 0.0% | 26.90% | 1.14% | 0.00% | 19.70% |
| 3 | 5.8% | 0.0% | 28.42% | 7.87% | 0.0% | 14.24% | 6.86% | 0.00% | 21.33% |
| 4 | 10.0% | 0.0% | <u>53.28%</u> | 14.20% | 0.0% | <u>53.99%</u> | <u>12.12%</u> | 0.00% | <u>53.64%</u> |
| 5 | 8.3% | 28.1% | - | 1.6% | <u>48.7%</u> | - | 4.9% | <u>38.4%</u> | - |
| 6 | 1.0% | 7.0% | - | 10.1% | 13.0% | - | 5.6% | 10.0% | - |
| 7 | 3.3% | <u>46.4%</u> | - | 7.1% | 26.1% | - | 5.2% | 36.3% | - |
| 8 | 7.7% | 14.6% | - | 5.0% | 8.2% | - | 6.3% | 11.4% | - |
| 9 | 10.4% | 0.0% | - | 2.7% | 0.0% | - | 6.5% | 0.0% | - |
| 10 | 4.5% | 0.0% | - | 6.9% | 0.0% | - | 5.7% | 0.0% | - |
| 11 | 1.7% | 0.0% | - | 1.1% | 0.0% | - | 1.4% | 0.0% | - |
| 12 | 13.0% | 0.0% | - | 9.2% | 0.0% | - | 11.1% | 0.0% | - |
| 13 | <u>13.5%</u> | 0.0% | - | 3.7% | 0.0% | - | 8.6% | 0.0% | - |
| 14 | 5.8% | - | - | 12.3% | - | - | 9.0% | - | - |
| 15 | 4.2% | - | - | 3.8% | - | - | 4.0% | - | - |
| 16 | 3.1% | - | - | 2.1% | - | - | 2.6% | - | - |
| 17 | 1.1% | - | - | 5.2% | - | - | 3.1% | - | - |
| 18 | 1.3% | - | - | 2.7% | - | - | 2.0% | - | - |
| 19 | 1.6% | - | - | 1.9% | - | - | 1.8% | - | - |
| 20 | 1.2% | - | - | 1.4% | - | - | 1.3% | - | - |

SW: stakeholder involvement weight; CW: criteria weight; Sce W: scenario weight.

**Table A2.** Results of pairwise comparison of the Saaty scale (from −9 to 9) for the Central Java group.

| NO | S3 | | | S4 | | | S5 | | | S6 | | | Average | | |
|---|---|---|---|---|---|---|---|---|---|---|---|---|---|---|---|
| | SW | CW | Sce W | SW | CW | Sce W | SW | CW | Sce W | SW | CW | Sce W | SW | CW | Sce W |
| 1 | 2.50% | 19.5% | 6.27% | 1.30% | 8.7% | 5.75% | 2.27% | <u>22.0%</u> | 5.79% | 3.00% | 13.1% | 5.74% | 2.27% | 15.84% | 5.89% |
| 2 | 2.33% | 0.0% | 13.86% | 1.00% | 26.6% | 14.24% | 2.92% | 0.0% | 12.43% | 2.28% | <u>54.0%</u> | 13.58% | 2.13% | 20.13% | 13.53% |
| 3 | 0.00% | 13.3% | 35.64% | 1.20% | 12.4% | 33.01% | <u>12.61%</u> | 0.0% | 51.62% | 1.56% | 0.0% | <u>45.25%</u> | 6.42% | 6.42% | 41.38% |
| 4 | <u>13.16%</u> | 6.7% | <u>44.23%</u> | 0.00% | 5.4% | <u>46.99%</u> | 12.60% | 5.3% | 30.16% | <u>13.95%</u> | 5.7% | 35.43% | 9.93% | 5.79% | <u>39.20%</u> |
| 5 | 3.3% | <u>44.4%</u> | - | 7.5% | <u>46.9%</u> | - | 0.0% | <u>38.4%</u> | - | 10.2% | 27.2% | - | 5.3% | <u>39.2%</u> | - |
| 6 | 8.1% | 0.0% | - | 1.5% | 0.0% | - | 7.5% | 0.0% | - | 0.0% | 0.0% | - | 4.3% | 0.0% | - |
| 7 | 7.2% | 0.0% | - | 4.4% | 0.0% | - | 7.4% | 0.0% | - | 4.0% | 0.0% | - | 5.7% | 0.0% | - |
| 8 | 13.0% | 4.6% | - | 3.1% | 0.0% | - | 11.0% | 21.7% | - | 4.1% | 0.0% | - | 7.8% | 6.6% | - |
| 9 | 3.3% | 0.0% | - | 10.9% | 0.0% | - | 4.1% | 0.0% | - | 9.8% | 0.0% | - | 7.0% | 0.0% | - |
| 10 | 10.6% | 0.0% | - | 1.6% | 0.0% | - | 7.2% | 0.0% | - | 5.7% | 0.0% | - | 6.3% | 0.0% | - |
| 11 | 4.4% | 0.0% | - | 5.9% | 0.0% | - | 5.7% | 0.0% | - | 7.4% | 0.0% | - | 5.9% | 0.0% | - |
| 12 | 10.5% | 11.6% | - | 3.0% | 0.0% | - | 10.6% | 0.0% | - | 1.0% | 0.0% | - | 6.3% | 2.9% | - |
| 13 | 4.5% | 0.0% | - | 10.3% | 0.0% | - | 3.2% | 12.6% | - | 7.9% | 0.0% | - | 6.5% | 3.1% | - |
| 14 | 8.7% | - | - | 2.2% | - | - | 4.2% | - | - | 12.4% | - | - | 6.9% | - | - |
| 15 | 1.1% | - | - | 12.6% | - | - | 1.3% | - | - | 5.3% | - | - | 5.1% | - | - |
| 16 | 1.1% | - | - | <u>12.9%</u> | - | - | 1.2% | - | - | 3.9% | - | - | 4.8% | - | - |
| 17 | 1.3% | - | - | 8.1% | - | - | 1.0% | - | - | 2.9% | - | - | 3.3% | - | - |
| 18 | 1.8% | - | - | 5.5% | - | - | 2.1% | - | - | 1.5% | - | - | 2.7% | - | - |
| 19 | 1.7% | - | - | 3.9% | - | - | 1.7% | - | - | 1.2% | - | - | 2.1% | - | - |
| 20 | 1.3% | - | - | 2.9% | - | - | 1.5% | - | - | 2.0% | - | - | 1.9% | - | - |

SW: stakeholder involvement weight; CW: criteria weight; Sce W: scenario weight.

**Table A3.** Results of pairwise comparison of the Saaty scale (from −9 to 9) for the Wonosobo group.

| NO | S7 | | | S8 | | | S9 | | | S10 | | | Average | | |
|---|---|---|---|---|---|---|---|---|---|---|---|---|---|---|---|
| | SW | CW | Sce W | SW | CW | Sce W | SW | CW | Sce W | SW | CW | Sce W | SW | CW | Sce W |
| 1 | 2.10% | 8.8% | 6.61% | 8.24% | 5.3% | 5.63% | 1.36% | 26.6% | 4.45% | 3.62% | 49.8% | 5.89% | 3.83% | 22.61% | 5.64% |
| 2 | 3.83% | 0.0% | 13.87% | 6.19% | 28.2% | 20.45% | 1.77% | 0.0% | 10.41% | 2.52% | 0.0% | 12.76% | 3.58% | 7.04% | 14.37% |
| 3 | 4.11% | 16.0% | 48.85% | 13.60% | 17.0% | 43.07% | 12.63% | 0.0% | 28.17% | 1.94% | 0.0% | 28.47% | 8.07% | 8.25% | 37.14% |
| 4 | 3.24% | 19.7% | 30.68% | 0.0% | 0.0% | 30.85% | 13.63% | 3.9% | 56.98% | 11.68% | 4.9% | 52.88% | 10.29% | 7.11% | 42.84% |
| 5 | 5.6% | 55.5% | - | 10.9% | 28.5% | - | 10.1% | 17.4% | - | 14.2% | 25.1% | - | 10.2% | 31.6% | - |
| 6 | 1.6% | 0.0% | - | 1.3% | 0.0% | - | 8.3% | 0.0% | - | 4.8% | 0.0% | - | 4.0% | 0.0% | - |
| 7 | 0.0% | 0.0% | - | 10.1% | 0.0% | - | 5.8% | 0.0% | - | 1.0% | 0.0% | - | 4.2% | 0.0% | - |
| 8 | 8.3% | 0.0% | - | 0.0% | 21.1% | - | 10.9% | 34.3% | - | 10.8% | 13.1% | - | 7.5% | 17.1% | - |
| 9 | 13.1% | 0.0% | - | 7.7% | 0.0% | - | 0.0% | 0.0% | - | 6.0% | 0.0% | - | 6.7% | 0.0% | - |
| 10 | 13.2% | 0.0% | - | 3.8% | 0.0% | - | 4.3% | 7.4% | - | 0.0% | 0.0% | - | 5.3% | 1.9% | - |
| 11 | 1.6% | 0.0% | - | 5.7% | 0.0% | - | 6.2% | 0.0% | - | 1.1% | 7.1% | - | 3.7% | 1.8% | - |
| 12 | 1.3% | 0.0% | - | 4.4% | 0.0% | - | 7.7% | 0.0% | - | 8.4% | 0.0% | - | 5.5% | 0.0% | - |
| 13 | 2.1% | 0.0% | - | 3.1% | 0.0% | - | 3.2% | 10.4% | - | 13.2% | 0.0% | - | 5.4% | 2.6% | - |
| 14 | 1.2% | - | - | 3.3% | - | - | 4.6% | - | - | 6.4% | - | - | 3.9% | - | - |
| 15 | 11.0% | - | - | 2.2% | - | - | 1.2% | - | - | 4.4% | - | - | 4.7% | - | - |
| 16 | 10.2% | - | - | 1.5% | - | - | 1.0% | - | - | 3.2% | - | - | 4.0% | - | - |
| 17 | 7.7% | - | - | 1.7% | - | - | 1.1% | - | - | 1.7% | - | - | 3.0% | - | - |
| 18 | 3.9% | - | - | 1.3% | - | - | 2.4% | - | - | 1.4% | - | - | 2.3% | - | - |
| 19 | 2.9% | - | - | 1.2% | - | - | 2.2% | - | - | 2.3% | - | - | 2.2% | - | - |
| 20 | 2.9% | - | - | 1.1% | - | - | 1.6% | - | - | 1.3% | - | - | 1.7% | - | - |

SW: stakeholder involvement weight; CW: criteria weight; Sce W: scenario weight.

**Table A4.** Results of pairwise comparison of the Saaty scale (from −9 to 9) for the Banjarnegara group.

| NO | S11 | | | S12 | | | S13 | | | S14 | | | Average | | |
|---|---|---|---|---|---|---|---|---|---|---|---|---|---|---|---|
| | SW | CW | Sce W | SW | CW | Sce W | SW | CW | Sce W | SW | CW | Sce W | SW | CW | Sce W |
| 1 | 10.22% | 14.1% | 5.89% | 7.59% | 7.4% | 5.72% | 1.69% | 8.3% | 5.82% | 6.00% | 8.2% | 5.70% | 6.37% | 9.51% | 5.78% |
| 2 | 3.28% | 26.1% | 14.81% | 4.06% | 26.1% | 13.62% | 1.56% | 0.0% | 19.88% | 3.01% | 25.7% | 12.23% | 2.98% | 19.51% | 15.14% |
| 3 | 11.04% | 7.4% | 44.27% | 13.56% | 14.1% | 32.26% | 12.56% | 0.0% | 43.93% | 7.47% | 0.0% | 30.07% | 11.16% | 5.38% | 37.63% |
| 4 | 13.73% | 48.3% | 35.03% | 12.57% | 0.0% | 48.39% | 13.55% | 4.2% | 30.36% | 10.71% | 4.2% | 52.00% | 12.64% | 14.18% | 41.45% |
| 5 | 12.7% | 0.0% | - | 10.0% | 48.3% | - | 10.0% | 46.5% | - | 8.0% | 47.9% | - | 10.2% | 35.7% | - |
| 6 | 3.1% | 0.0% | - | 10.8% | 0.0% | - | 7.6% | 0.0% | - | 3.2% | 0.0% | - | 6.2% | 0.0% | - |
| 7 | 4.4% | 0.0% | - | 4.4% | 0.0% | - | 4.3% | 0.0% | - | 5.5% | 0.0% | - | 4.6% | 0.0% | - |
| 8 | 7.9% | 4.0% | - | 3.1% | 4.0% | - | 6.0% | 26.4% | - | 9.9% | 0.0% | - | 6.7% | 8.6% | - |
| 9 | 5.9% | 0.0% | - | 2.9% | 0.0% | - | 3.3% | 0.0% | - | 4.2% | 0.0% | - | 4.1% | 0.0% | - |
| 10 | 2.3% | 0.0% | - | 2.3% | 0.0% | - | 4.1% | 0.0% | - | 1.3% | 0.0% | - | 2.5% | 0.0% | - |
| 11 | 0.0% | 0.0% | - | 8.2% | 0.0% | - | 10.8% | 0.0% | - | 1.5% | 13.9% | - | 5.1% | 3.5% | - |
| 12 | 8.5% | 0.0% | - | 0.0% | 0.0% | - | 8.2% | 0.0% | - | 13.5% | 0.0% | - | 7.5% | 0.0% | - |
| 13 | 6.4% | 0.0% | - | 6.0% | 0.0% | - | 0.0% | 14.6% | - | 1.1% | 0.0% | - | 3.4% | 3.7% | - |
| 14 | 2.4% | - | - | 5.6% | - | - | 5.6% | - | - | 0.0% | - | - | 3.4% | - | - |
| 15 | 1.8% | - | - | 2.1% | - | - | 1.3% | - | - | 3.2% | - | - | 2.1% | - | - |
| 16 | 1.7% | - | - | 1.6% | - | - | 1.2% | - | - | 12.5% | - | - | 4.2% | - | - |
| 17 | 1.4% | - | - | 1.5% | - | - | 1.0% | - | - | 3.9% | - | - | 1.9% | - | - |
| 18 | 1.3% | - | - | 1.3% | - | - | 2.9% | - | - | 1.0% | - | - | 1.6% | - | - |
| 19 | 1.1% | - | - | 1.2% | - | - | 2.3% | - | - | 1.2% | - | - | 1.4% | - | - |
| 20 | 1.0% | - | - | 1.1% | - | - | 2.1% | - | - | 3.0% | - | - | 1.8% | - | - |

SW: stakeholder involvement weight; CW: criteria weight; Sce W: scenario weight.

**Table A5.** Results of pairwise comparison of the Saaty scale (from −9 to 9) for the local community group.

| NO | S15 | | | S16 | | | S17 | | | Average | | |
|---|---|---|---|---|---|---|---|---|---|---|---|---|
| | SW | CW | Sce W | SW | CW | Sce W | SW | CW | Sce W | SW | CW | Sce W |
| 1 | 10.02% | 3.2% | 27.35% | 12.61% | 3.4% | 12.42% | 14.58% | 35.3% | 53.35% | 12.40% | 13.95% | 31.04% |
| 2 | 8.17% | 25.0% | 21.71% | 10.04% | 41.9% | 30.28% | 12.04% | 0.0% | 27.47% | 10.08% | 22.29% | 26.49% |
| 3 | 10.83% | 5.0% | 31.37% | 13.60% | 5.2% | 36.44% | 11.15% | 17.2% | 12.01% | 11.86% | 9.16% | 26.60% |
| 4 | 13.56% | 0.0% | 19.57% | 10.85% | 0.0% | 20.86% | 9.34% | 7.1% | 7.16% | 11.25% | 2.36% | 15.87% |
| 5 | 12.6% | 14.9% | - | 7.8% | 14.7% | - | 7.0% | 0.0% | - | 9.1% | 9.8% | - |
| 6 | 2.9% | 0.0% | - | 4.1% | 0.0% | - | 3.4% | 0.0% | - | 3.5% | 0.0% | - |
| 7 | 1.7% | 0.0% | - | 4.4% | 0.0% | - | 1.8% | 0.0% | - | 2.6% | 0.0% | - |
| 8 | 2.3% | 0.0% | - | 2.3% | 0.0% | - | 2.6% | 0.0% | - | 2.4% | 0.0% | - |
| 9 | 3.1% | 43.3% | - | 3.1% | 26.4% | - | 2.5% | 40.4% | - | 2.9% | 36.7% | - |
| 10 | 2.1% | 0.0% | - | 2.1% | 0.0% | - | 1.9% | 0.0% | - | 2.1% | 0.0% | - |
| 11 | 4.1% | 0.0% | - | 7.7% | 0.0% | - | 3.7% | 0.0% | - | 5.2% | 0.0% | - |
| 12 | 4.4% | 0.0% | - | 5.6% | 0.0% | - | 5.1% | 0.0% | - | 5.0% | 0.0% | - |
| 13 | 5.6% | 8.6% | - | 6.0% | 8.4% | - | 4.7% | 0.0% | - | 5.5% | 5.7% | - |
| 14 | 1.6% | - | - | 2.9% | - | - | 1.5% | - | - | 2.0% | - | - |
| 15 | 0.0% | - | - | 1.7% | - | - | 8.7% | - | - | 3.5% | - | - |
| 16 | 7.6% | - | - | 0.0% | - | - | 6.5% | - | - | 4.7% | - | - |
| 17 | 6.0% | - | - | 1.6% | - | - | 0.0% | - | - | 2.5% | - | - |
| 18 | 1.2% | - | - | 1.3% | - | - | 1.3% | - | - | 1.3% | - | - |
| 19 | 1.3% | - | - | 1.3% | - | - | 1.2% | - | - | 1.3% | - | - |
| 20 | 1.0% | - | - | 1.0% | - | - | 1.1% | - | - | 1.0% | - | - |

SW: stakeholder involvement weight; CW: criteria weight; Sce W: scenario weight.

**Table A6.** Results of pairwise comparison of the Saaty scale (from −9 to 9) for the user group.

| NO | S18 | | | S19 | | | S20 | | | Average | | |
|---|---|---|---|---|---|---|---|---|---|---|---|---|
| | SW | CW | Sce W | SW | CW | Sce W | SW | CW | Sce W | S | C | Sce |
| 1 | 1.42% | 9.4% | 5.73% | 1.01% | 4.4% | 6.57% | 1.73% | 26.3% | 6.17% | 1.39% | 13.4% | 6.15% |
| 2 | 1.31% | 41.0% | 13.70% | 1.56% | 27.2% | 14.47% | 2.14% | 0.0% | 12.76% | 1.67% | 22.7% | 13.64% |
| 3 | 9.21% | 5.5% | 37.47% | 1.21% | 9.0% | 40.31% | 1.98% | 0.0% | 50.83% | 4.14% | 4.8% | 42.87% |
| 4 | 3.34% | 3.1% | 43.10% | 2.11% | 0.0% | 38.65% | 3.06% | 5.7% | 30.25% | 2.84% | 2.9% | 37.34% |
| 5 | 2.7% | 24.8% | - | 1.7% | 44.9% | - | 1.5% | 12.2% | - | 2.0% | 27.3% | - |
| 6 | 1.8% | 0.0% | - | 3.1% | 0.0% | - | 1.2% | 0.0% | - | 2.0% | 0.0% | - |
| 7 | 4.8% | 0.0% | - | 7.6% | 0.0% | - | 4.0% | 0.0% | - | 5.5% | 0.0% | - |
| 8 | 7.0% | 0.0% | - | 8.2% | 14.6% | - | 5.5% | 55.8% | - | 6.9% | 23.5% | - |
| 9 | 11.4% | 0.0% | - | 12.6% | 0.0% | - | 11.9% | 0.0% | - | 12.0% | 0.0% | - |
| 10 | 14.5% | 16.2% | - | 13.6% | 0.0% | - | 12.8% | 0.0% | - | 13.6% | 5.4% | - |
| 11 | 1.1% | 0.0% | - | 1.3% | 0.0% | - | 10.0% | 0.0% | - | 4.1% | 0.0% | - |
| 12 | 3.5% | 0.0% | - | 6.0% | 0.0% | - | 7.5% | 0.0% | - | 5.7% | 0.0% | - |
| 13 | 8.5% | 0.0% | - | 10.0% | 0.0% | - | 5.3% | 0.0% | - | 8.0% | 0.0% | - |
| 14 | 11.3% | - | - | 10.8% | - | - | 5.7% | - | - | 9.3% | - | - |
| 15 | 1.8% | - | - | 2.9% | - | - | 1.0% | - | - | 1.9% | - | - |
| 16 | 3.3% | - | - | 5.6% | - | - | 2.7% | - | - | 3.9% | - | - |
| 17 | 2.4% | - | - | 4.1% | - | - | 1.6% | - | - | 2.7% | - | - |
| 18 | 0.0% | - | - | 4.4% | - | - | 10.5% | - | - | 5.0% | - | - |
| 19 | 6.2% | - | - | 0.0% | - | - | 9.7% | - | - | 5.3% | - | - |
| 20 | 4.5% | - | - | 2.3% | - | - | 0.0% | - | - | 2.3% | - | - |

SW: stakeholder involvement weight; CW: criteria weight; Sce W: scenario weight.

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
