# Peer review of "Development of Multi-Actor Multi-Criteria Analysis Based on the Weight of Stakeholder Involvement in the Assessment of Natural–Cultural Tourism Area Transportation Policies"

_algorithms, doi:10.3390/a14070217_

Round 1
Reviewer 1 Report
The authors address an interesting and up-to-date topic developing a decision-making framework to deal with a large number of stakeholders. The approach is interesting and based on a widely accepted approach as the basic idea is to make the current decision-making environment more resilient and efficient. Although the topic is interesting, several major insufficiencies need to be improved. These insufficiencies can be summed up in poor scientific writing and a lack of clarity.
Suggestions for improvement:
- The manuscript should be set according to the Journal’s template and instruction to authors (text, figures, tables, equations, references, abbreviations, etc.). .). Please define abbreviations in the text before their first use, variables, SI units, figures, etc. For example, the figures are unreadable and/or not connected with text, missing references, etc.
- Check and improve the English language and grammar throughout the paper (check misspellings, writing in the first person, etc.) and all figures and tables (both must be readable). The authors should be consistent in writing
- Although the title is informative, the abstract fails to bring any novelty or uniqueness. Overall, the abstract is poorly written with no clear structure of the used methods and approaches, findings, and conclusions
- The introduction does not provide sufficient background or includes all relevant references. The used references are not novel nor based on previous scientific papers. Also, some fundamental references are missing as well as the recent ones considering the research problem. Also, the authors newer addressed other approaches of dealing with multi-stakeholders throughout the whole manuscript. The research problem is not clear while the research goals and hypotheses are not clearly stated
- The literature review should be improved. At the moment it lacks a critical overview of the other approaches in solving the stated research problem and the methodology upgrade that is proposed by this research
- The research design is not clearly written. The research methodology should be clear and the hows and the whys of used methods should be clearly visible. The validation is especially important, therefore authors are urged to give insight into the validation process. Also, add some additional discussion of findings concerning the research framework as well as research goals and hypotheses are needed
- The authors are urged to draw more specific conclusions. At the moment it seems like good observations and arguments are currently missing from the discussion section. There should be a clear connection with the research problem, goals, and results
Overall, I strongly urge the authors to reconsider the above-mentioned comments, rewrite the paper accordingly, and resubmit. Therefore at the moment, the manuscript does not reach the desired level for publishing.
Author Response
Based on your review, we have made improvements in this manuscript, hope you like it:
1.The abstract has been fixed
2.Introduction has been fixed
3. References have used previous and latest research.
4. Literature review has been improved.
5. Methodology has been improved
6. Conclusions have been updated and are more specific.
7.Observations and arguments have been added
I hope you like it and thank you.
regards,
Titus Hari Setiawan
Reviewer 2 Report
The paper presents a solid application of the well-known MAMCA method created by Macharis, Its novelty lies in the new application area of natural-cultural tourism areas and the created MAMCA model for this decision problem.
From a decision theory point of view, I only lack one issue. We have to make a clear distinction between negotiable and non-negotiable problems and surveys in multi-criteria group decision-making. MAMCA is restricted to negotiable MCDM cases. However, in case the pattern is large-scale or one or more participant groups do not intend or cannot be involved in a negotiation process, non-negotiable methods are required. Please see and refer Amenda et al (2021), On the choice of weights for aggregating judgements in non-negotiable group decision-making, EJOR, 288(1); and Ghorbanzadeh et al (2019), Sustainable urban transport planning considering different stakeholder groups by an Interval-AHP decision support model.
By these references, the research will be clearly outlined and the paper can be published.
Author Response
In this study, an external assessment is carried out on the involvement of stakeholders in making transportation policy decisions in natural-cultural tourism protected areas which result in the weight of stakeholder involvement. Stakeholder weighting is carried out by peer assessment and by using the pairwise comparison method. In this study, the general AHP was used instead of the interval AHP (Ghorbanzadeh, Moslem, Blaschke, & Duleba, 2018), considering that the evaluators consisted of 20 stakeholders who could be grouped into 6 groups. These groups are representatives from the central government, provincial governments, local governments, local communities, and users/tourists. They manage and are related to 4 main problems, namely transportation, tourism, the environment, and historical heritage which lead to the welfare of local residents and actors providing transportation and tourism services. According to them, these problems are negotiable so that the writer feels there is no need to formulate a method or take into account the existence of non-negotiable things (Amenta, Lucadamo, & Marcarelli, 2021). Stakeholders, in the field, are parties who continue to negotiate with each other so that their interests can be realized. Heterogeneity or multi-stakeholder and multi-criteria are issues that need to be considered in determining policies regarding sustainable tourism destinations (Lindberg, Fitchett, & Martin, 2019). Finally, stakeholder participation in multi-actor multi-criteria analysis with the weight of stakeholder method produces a transportation policy scenario for our case study. thank you.
regards
Titus Hari Setiawan
Round 2
Reviewer 1 Report
In the revised version authors gave some additional insights into their research and also acted upon given comments and suggestions but still fail to give additional value to the manuscript. Supplementary material is a good addition and gives necessary clarifications but still, the discussion section lacks validation and placing the findings in relation to the research framework as well as research goals and hypotheses. By the way, they are still not clearly stated. The literature review still lacks a critical overview of the other approaches in solving the stated research problem and the methodology upgrade that is proposed by this research. For example, there is no mention of Jajac, Marović, Deluka, or Waaub who published on the MCDA in a multi-stakeholder environment especially in transportation. I strongly urge the authors to reconsider the prior comments, review cycle, and make the changes accordingly. The major problem lies in the poor writing style, which leads to many inconsistencies and consequently is difficult to follow. Therefore, the authors are encouraged to work on and improve their scientific writing and give all necessary aspects of scientific written communication.
Author Response
Dear reviewer 1
We have made improvements to the introduction and added literature on the MCDA field with multi-stakeholder engagement.
Thank you.
regerds
Titus Hari Setiawan

Round 3
Reviewer 1 Report
In the revised version authors gave additional insights into their research but still fail to give a clear insight into research goals and hypotheses. All arguments are already written but also there should be a clear connection with the research problem, goals, results, and conclusion. At the moment there is a connection between the research problem, used methods, and conclusions. The authors are advised to add the goals as well. With them, the whole manuscript will be clearer. Overall, I believe that the article provides valuable content to the present body-of-knowledge especially for MAMCA users, and after these minor suggestions, it deserves to be published.
Author Response
Based on reviewer 1 round 3, we add the goals of this paper at the end of the introduction and in the conclusion section (in red). Thank you.
